# The Influence of Preoperative Anticoagulant and Antiplatelet Therapy on Rebleeding Rates in Patients Suffering from Spinal Metastatic Cancer: A Retrospective Cohort Study

**DOI:** 10.3390/cancers16112052

**Published:** 2024-05-29

**Authors:** Christoph Orban, Anto Abramovic, Raphael Gmeiner, Sara Lener, Matthias Demetz, Claudius Thomé

**Affiliations:** Department of Neurosurgery, Medical University Innsbruck, 6020 Innsbruck, Austria; christoph.orban@tirol-kliniken.at (C.O.); claudius.thome@tirol-kliniken.at (C.T.)

**Keywords:** spinal metastasis, surgery, anticoagulants, antiplatelet therapy, bleeding risk, neurological deficits, coagulation management, rebleeding, elderly patients, perioperative care

## Abstract

**Simple Summary:**

The study investigates the impact of preoperative anticoagulant and antiplatelet use on bleeding risks in patients undergoing surgery for spinal metastases, particularly focusing on the elderly population. Due to advancements in cancer treatments, more patients over 65 require such surgeries to address neurological deficits and spinal instability. Conducted retrospectively from 2010 to 2023, the study analyzed 290 patients’ data, including demographics, neurological status, surgical details, anticoagulant use, and coagulation management. Among the patients, 24.1% were on anticoagulants or antiplatelets preoperatively, and the rebleeding rate within 30 days was 4.5%, which was not significantly related to anticoagulant use. A significant correlation was found between preoperative neurological deficits and rebleeding risk, and fewer surgical levels treated correlated with higher postoperative bleeding. The study concludes that surgery for spinal metastases is generally safe regardless of anticoagulation status, but it emphasizes the need for individualized preoperative planning and risk assessment to optimize patient outcomes.

**Abstract:**

Introduction: The age of patients requiring surgery for spinal metastasis, primarily those over 65, has risen due to improved cancer treatments. Surgical intervention targets acute neurological deficits and instability. Anticoagulants are increasingly used, especially in the elderly, but pose challenges in managing bleeding complications. The study examines the correlation between preoperative anticoagulant/antiplatelet use and bleeding risks in spinal metastasis surgery, which is crucial for optimizing patient outcomes. Material and Methods: In a retrospective study at our department from 2010 to 2023, spinal tumor surgery patients were analyzed. Data included demographics, neurological status, surgical procedure, preoperative anticoagulant/antiplatelet use, intra-/postoperative coagulation management, and the incidence of rebleeding. Coagulation management involved blood loss assessment, coagulation factor administration, and fluid balance monitoring post-surgery. Lab parameters were documented at admission, preop, postop, and discharge. Results: A cohort of 290 patients underwent surgical treatment for spinal metastases, predominantly males (63.8%, *n* = 185) with a median age of 65 years. Preoperatively, 24.1% (*n* = 70) were on oral anticoagulants or antiplatelet therapy. Within 30 days, a rebleeding rate of 4.5% (*n* = 9) occurred, unrelated to preoperative anticoagulation status (*p* > 0.05). A correlation was found between preoperative neurologic deficits (*p* = 0.004) and rebleeding risk and the number of levels treated surgically, with fewer levels associated with a higher incidence of postoperative bleeding (*p* < 0.01). Conclusions: Surgical intervention for spinal metastatic cancer appears to be safe regardless of the patient’s preoperative anticoagulation status. However, it remains imperative to customize preoperative planning and preparation for each patient, emphasizing meticulous risk–benefit analysis and optimizing perioperative care.

## 1. Introduction

The age of patients undergoing surgery for spinal metastases has shown a consistent increase over time, predominantly affecting those over 65 years old [1,2]. This shift can be attributed to advancements in cancer treatment, including novel options like targeted therapies and immune checkpoint inhibitors, which have extended the life expectancy of cancer patients [3,4]. Osseous spinal metastases, the most common form, often lead to significant morbidity. Surgical intervention is typically reserved for cases involving acute neurological deficits, substantial instability causing deficits, or unmanageable pain, particularly in patients with a favorable short-term prognosis [5]. However, determining the appropriate indication and extent of surgical treatment necessitates a meticulous risk–benefit analysis.

The use of oral anticoagulants and antiplatelet therapy has become increasingly common for preventing thromboembolic events, particularly in the elderly. Yet, managing patients who experience bleeding complications or require emergency surgery remains a topic of ongoing debate. While many spinal procedures are elective, spine surgeons frequently encounter emergencies such as compressive spinal metastatic cancer, resulting in neurological impairment necessitating immediate decompression to prevent permanent damage. In patients on antithrombotic or antiplatelet medication, impaired hemostasis presents a risk of intraoperative and postoperative bleeding complications with potentially life-threatening implications [6]. Thus, careful consideration and optimized strategies for managing hemostasis are mandatory in such scenarios.

The objective of this study was to assess the correlation between the preoperative administration of oral anticoagulation or antiplatelet drugs and the risk of intra- and postoperative bleeding and complications in patients undergoing (acute) surgery for spinal metastatic cancer.

## 2. Material and Methods

We conducted a retrospective study involving patients with spinal tumors who underwent spinal surgery at the authors’ department between 2010 and 2023. The inclusion criteria were narrowed to exclude patients with central nervous system tumors, spinal meningiomas, and vascular malformations. The variables collected for analysis included demographics, American Spinal Injury Association (ASIA) and Medical Research Council (MRC) scores upon admission, tumor type, surgical procedure, details of preoperative anticoagulant intake, intraoperative and postoperative coagulation management, duration of surgery, unplanned return to the operation room within 30 days, and incidence of rebleeding. Outcomes were evaluated based on the rebleeding (defined as clinically significant progressive pain and/or neurological deficit, requiring revision surgery) and complication rate, MRC/ASIA scores at discharge, and mortality within one year after surgery.

Surgical procedures were categorized into decompression, instrumentation, corpectomy, biopsy, and vertebroplasty, with the most invasive procedure being selected for each case in the analysis. Detailed preoperative anticoagulation data, including the type of medication, the duration between the last intake of anticoagulants and surgery, and the reversion of anticoagulation, were recorded. The management of coagulation both intraoperatively and postoperatively involved assessing blood loss, administering coagulation factors or blood products, and monitoring fluid balance and wound drainage within the first 72 h post-surgery. Laboratory parameters, such as hemoglobin, thrombocytes, and coagulation parameters including International Normalized Ratio (INR), fibrinogen, and Quick’s test, were documented upon admission, 24 h preoperatively, 24 h postoperatively, and upon discharge.

## 3. Statistical Analysis

The statistical analysis of the data was conducted using IBM SPSS Statistics Version (25.0.1, Armonk, NY, USA) and Microsoft Excel Version 1090, Microsoft Corporation, Redmond (Washington, DC, USA). We established statistical significance at a *p*-value of less than 0.05. For the analysis of demographic data, we applied the Kolmogorov–Smirnov test for interval-scaled data and the *t*-test or Mann–Whitney U test as appropriate. The Chi-square test was used for categorical or ordinal data. To assess risk factors, we utilized a binomial logistic regression analysis with pre-defined models and conducted a Bonferroni post hoc analysis to adjust for multiple comparisons. Changes from preoperative to postoperative states were evaluated using the Wilcoxon signed-rank test. Results were visually represented in tables and graphs using Microsoft Excel (Version 1090).

## 4. Results

Those included numbered 290 patients who underwent surgical treatment for spinal tumors at the authors’ institution. The cohort consisted predominantly of males (185 patients; 63.8%), with a median age of 65 years (interquartile range [IQR] 57; 73 years), and a median BMI of 23.8 (IQR 21.2; 27). The median preoperative ASA score was 3 (IQR 2–3), indicating moderate to severe systemic disease in the patient population. The distribution of preoperative risk factors is shown in Table 1. The median duration of the surgical procedures was 156 min (IQR 102; 250 min), and the median loss of blood was 400 mL (IQR 100, 900 mL). The surgical interventions included vertebroplasty (25 cases, 8.6%), biopsy (2 cases, 0.7%), corpectomy (50 cases, 17.2%), instrumentation (96 cases, 33.1%), and decompression (116 cases, 40%). The most frequent tumor entities were prostate cancer (21.4%, 62 patients), non-small cell lung cancer (NSCLC, 16.2%, 47 patients), and myeloma (10.3%, 30 patients). Preoperative neurologic deficits were present in 53.1% of patients (154 patients).

Regarding anticoagulation status, 24.1% of patients (70 patients) were on oral anticoagulant therapy preoperatively (Table 2), with preoperative embolization performed in 3.1% (9 patients) and the reversal of anticoagulation in 1.4% (4 patients).

The surgical data revealed that 16.9% of the operations (49 cases) were executed urgently, categorized as surgeries conducted within 24 h following the onset of symptoms. During surgery, 32.8% of the cases (95 patients) necessitated the intraoperative administration of coagulation factors. This practice showed a significant association with preoperative INR levels (*p* = 0.004) and Quick’s test results (*p* < 0.01). The intraoperative provision of coagulation factors was predominantly observed in surgeries of extended duration (*p* < 0.01) and in more complex procedures, characterized as involving multiple levels (*p* = 0.015). Intraoperative complications occurred in 4.8% of the cases (14 patients), as detailed in Table 3. The various levels of surgery are documented in Table 4.

Rebleeding rate amounted to 4.5% (9 patients) within 30 days and the mortality rate was 44.8% (130 patients) within the first postoperative year. Blood tests including thrombocyte count, hemoglobin (Hb) levels, Quick’s test, and International Normalized Ratio (INR) did not demonstrate a significant impact on the emergence of postoperative rebleeding (>0.05). We utilized the Chi-Square test to examine the influence of various known perioperative risk factors on postoperative rebleeding. The factors analyzed are shown in Table 5. Neither patients with hepatopathy (0.065) nor nephropathy (0.857) harbored an increased risk for postoperative bleeding.

Despite the overall non-significance of many factors, two significant predictors of postoperative rebleeding were identified. A preoperative neurologic deficit was associated with an increased risk of rebleeding (*p* = 0.004). Additionally, the level of surgery (ranging from 1 level to 10 levels) showed a significant influence, with a higher incidence of postoperative rebleeding observed in surgeries involving fewer levels (*p* < 0.001).

Preoperatively, the distribution of patients according to the ASIA scale was as follows: ASIA A in 10 patients (3%), ASIA B in 9 patients (3%), ASIA C in 37 patients (13%), ASIA D in 85 patients (29%), and ASIA E in 144 patients (50%). At the time of discharge, there was a noticeable shift in this distribution: ASIA A in 5 patients (2%), ASIA B in 8 patients (3%), ASIA C in 24 patients (8%), ASIA D in 69 patients (24%), and ASIA E in 162 patients (56%). This change represented a statistically significant improvement in neurological status (*p* < 0.001) (Figure 1). Earlier surgery did not show significantly better improvement in the pre- to postoperative ASIA scale (0.644). Regarding muscle strength as measured with the MRC scale, the preoperative grades were observed as follows (0–5): 16 patients (6%) with the lowest grade, followed by 12 patients (4%), 27 patients (9%), 33 patients (11%), 49 patients (17%), and the majority, 149 patients (51%), had MRC 5. Upon discharge, the grading showed improvement: grades ranging from 0 to 5 were observed in 9 patients (3%), 7 patients (2%), 17 patients (6%), 33 patients (11%), 45 patients (16%), and most notably, 166 patients (57%) scored the highest grade, respectively (Figure 2). This progression also demonstrated a significant improvement from preoperative to postoperative status (*p* ≤ 0.001).

## 5. Discussion

We present the findings of our retrospective study aimed at investigating risk factors for postoperative hemorrhage in the surgical treatment of spinal metastatic cancer. Our analysis revealed that preoperative anticoagulant or antiplatelet therapy does not correlate with an increased rebleeding rate. However, it was observed that preoperative neurological deficits and a smaller number of operated levels are associated with a higher incidence of postoperative hemorrhage.

In terms of age, gender distribution, and tumor entity, the demographic data in our study are comparable to others and did not exhibit any significant differences regarding perioperative complications [7,8]. Various modifiable and non-modifiable risk factors have been previously assessed for rebleeding after metastatic spinal surgery. Perioperative risk factors such as smoking status, diabetes, and kidney or liver disease, which have shown no significance in terms of increased bleeding rates, were also insignificant in our cohort [9,10,11,12,13,14,15,16,17].

Elderly individuals often take anticoagulant or antiplatelet medication, yet the management of these patients who might require emergency surgery, such as operative treatment for spinal metastatic cancer, remains a topic of ongoing debate [18,19,20].

While standardized guidelines exist for discontinuing these drugs, the data on urgent cases are sparse. For instance, patients with coronary heart disease on aspirin or dual therapy with clopidogrel face higher intraoperative bleeding risks. However, a cohort study by Cuellar et al. found no increased bleeding complications in surgeries with aspirin alone [21]. Despite the rise in direct oral anticoagulant (DOAC) use, specific perioperative guidelines for spine surgery are lacking. Dabigatran shows similar perioperative bleeding and thrombotic risks compared to warfarin, with procedures often feasible within 48 h of discontinuation [22]. The short preoperative discontinuation (<24 h) of DOACs may be acceptable, albeit with a slightly increased need for perioperative PRBC transfusion [23]. A careful risk assessment is essential before surgery, especially for those with a more complex history of antiplatelet/anticoagulation therapy. None of the patients in the discussed groups were on dual antiplatelet therapy. The overall rebleeding rate in our cohort was 4.5%, congruent with the existing literature [24,25]. There was no statistically significant correlation between the preoperative intake of anticoagulation and/or antiplatelet therapy and the risk for rebleeding or general perioperative complications. Similarly, no significant correlation between the duration of medication pause and postoperative bleeding was observed. This implies that acute surgery for spinal metastatic cancer in patients on anticoagulant/antiplatelet therapy seems to be a safe option to preserve function.

The literature provides data on the reversal of anticoagulants and the associated reduction in bleeding during surgery [14,23,26,27,28]. However, our study did not show a significant reduction in the postoperative bleeding rate after the application of factors, which was performed in 1.4% of our patient cohort. Nonetheless, there might be some advantages in practice, always considering the risk of iatrogenic thromboembolic events and other associated complications [29,30]. The same applies to the intraoperative administration of coagulation factors, as 32.8% of patients in our cohort received some form of coagulation factor during surgery. Nevertheless, no thromboembolic complications occurred in our cohort. Based on these results, it should be discussed whether the timing and administration of factors should be optimized perioperatively. Laboratory markers such as hemoglobin, thrombocytes, Quick’s test, and INR are described for the better prediction of perioperative risks, such as (re)bleeding or the need for perioperative blood transfusions [30,31,32,33]. However, those markers were also not significantly correlated with the risk of rebleeding after surgery for metastatic spinal cancer in our cohort. Still, these parameters could help identify patients at higher perioperative bleeding risk and should be interpreted carefully.

We aimed to evaluate all surgical methods employed for the treatment of spinal metastases and assess their associated risks of bleeding or rebleeding depending on the intake of anticoagulant medication. Consequently, our analysis did not restrict itself to any specific surgical technique. Our objective was to identify statistically significant differences in postoperative bleeding risks among the different surgical methods. Overall, the complexity of the surgical procedure is associated with a higher risk of extended intraoperative blood loss [34]. Nonetheless, a single-stage procedure with stabilization and decompression can preserve the patient’s functional status and increase the chances of adjuvant therapy, thereby extending life expectancy [35]. In our study, a correlation was observed between operations performed at fewer levels and an increased risk of postoperative bleeding. Specifically, among 107 decompression surgeries (median 2 levels; IQR 1, 3), there were a total of seven (6.5%) postoperative hemorrhages. In contrast, the risk of rebleeding was significantly lower in patients receiving more complex treatment for more levels (median 3; IQR 2, 4 levels). One reason for this trend could be that patients undergoing decompression were only not consistently monitored in an intensive care unit post-surgery, and the inadequate management of postoperative blood pressure could contribute to the higher incidence of bleeding observed in this patient group [36]. Additionally, the theory of smaller compartments and less possibility to compensate for the increase in volume triggered by hemorrhage, leading to clinically apparent rebleeding and neurological deterioration, could be possible [37] as well as less aggressive tumor resection in these cases, leaving more pathologic tissue prone to hemorrhage behind. It could also be speculated that simply the awareness of rebleeding risk was lower in these “smaller” surgeries. Therefore, this unexpectedly high rebleeding rate in short-segment procedures is an important finding of our study.

As demonstrated in our study, the preoperative neurological status correlates with postoperative outcomes. This has also been described by Laufer et al. Reasons for this include the potential of rehabilitation, mobilization, and also psychological factors [38]. Additionally, rebleeding may become symptomatic earlier in patients with preoperative deficits. Fundamentally, optimal patient treatment must consider various factors such as neurological, oncological, mechanical, and systemic status. Maximum tumor resection often entails increased surgery-related morbidity [39,40,41]. The goal should be to address acute neurological symptoms and separate the tumor mass sufficiently from neural structures to allow further treatment [42,43].

Altogether, the risk of early surgery in elderly patients suffering from spinal metastatic cancer, irrespective of their anticoagulation/antiplatelet status, does not appear to be increased and does not justify a delay in patients with acute neurologic symptoms. However, it seems to be relevant to closely monitor this patient group and, if necessary, to follow them up in an intensive care unit to enable optimal management depending on their individual needs, also focusing on the postoperative admission of anticoagulative agents, as patients suffering from cancer knowingly carry a high risk for thromboembolic events.

Limitations of this study include its retrospective design and the fact that this patient cohort was treated at a single institute. Proper blood management may have overcome or masked a significant impact of preoperative anticoagulation/antiplatelet therapy. Additionally, the decision regarding perioperative blood product substitution was not solely made by the surgical team. Thus, blood product substitution may have occurred at peripheral hospitals or other departments without being documented in our records. Additionally, data collection changes over a number of years in management may have also influenced the results, like different anticoagulation medications or the introduction of minimally invasive surgery [44].

## 6. Conclusions

In conclusion, the surgical management of spinal metastatic cancer poses challenges, especially with increasing life expectancy and anticoagulant/antiplatelet use. While our study found no significant link between (re)bleeding and preoperative medication, personalized preoperative planning remains essential, highlighting the need for further research to optimize treatment strategies.

## Figures and Tables

**Figure 1 cancers-16-02052-f001:**
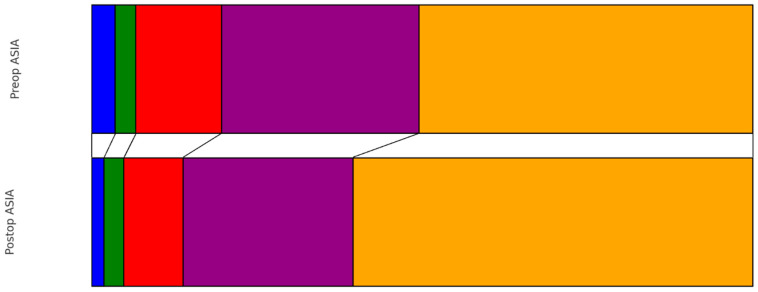
Pre- vs. Postoperative distribution of ASIA scale (blue: ASIA A, green: ASIA B, red: ASIA C, purple: ASIA D, orange: ASIA E).

**Figure 2 cancers-16-02052-f002:**
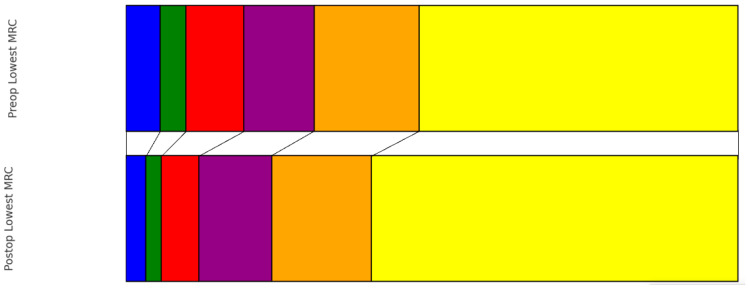
Pre- vs. Postoperative distribution of lowest MRC score (blue: MRC 0, green: MRC 1, red: MRC 2, purple: MRC 3, orange: MRC 4, yellow: MRC 5).

**Table 1 cancers-16-02052-t001:** Preoperative Risk Factors for the occurrence of postoperative bleeding.

Risk Factor	Number (%)
Smoking history	57 (19.7)
Hypertension	123 (42.4)
Diabetes mellitus	31 (10.7)
Coronary heart disease	41 (14.1)
Hepatopathy	24 (8.3)
Nephropathy	56 (19.3)
Coagulation disorder	1 (0.3)

**Table 2 cancers-16-02052-t002:** Variation in the Use of Various Anticoagulants Across the Patient Cohort.

Anticoagulant Intake	Number (%)
Apixaban	8 (2.8)
Heparin	1 (0.3)
Edoxaban	1 (0.3)
Clopidogrel	2 (0.7)
Acenocumarol	5 (1.7)
Acetylsalicylic acid	45 (15.5)
Rivaroxaban	3 (1)
Multiple	1 (0.3)
Unknown	4 (1.3)

**Table 3 cancers-16-02052-t003:** Occurrence and type of perioperative complications.

Perioperative Complication	Number (%)
Incidental durotomy	11 (3.8)
Cement leakage	1 (0.3)
Wound healing disorder	2 (0.7)

**Table 4 cancers-16-02052-t004:** Distribution of the level of surgery.

Surgery Level	Number (%)
Cervical	39 (13.4)
Thoracic	160 (55.2)
Lumbar	76 (26.2)
Sacral	2 (0.7)
Multiregional	13 (4.5)

**Table 5 cancers-16-02052-t005:** Comparison of perioperative factors in patients with vs. without postoperative bleeding.

	Rebleeding	No Rebleeding	*p*-Value
Preop reversion of anticoagulants	1 (11.1)	3 (1.1)	0.121
Administration of coagulation factors			
Preop	0	9 (3.2)	0.581
Intraop	4 (44.4)	90 (32.5)	0.492
Postop	1 (11.1)	27 (9.7)	0.913
Intraoperative complications	0	13 (4.7)	0.503
Use of drainage	7 (77.8)	169 (61.0)	0.491
Preoperative embolization	0	9 (3.2)	0.582
Anticoagulant intake	3 (33.3)	67 (24.2)	0.530
Duration of surgery (min.; median; IQR)	153 (82, 234)	157 (104, 254)	0.625

## Data Availability

The data presented in this study are available on request from the corresponding author.

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
