# Peer review of "The Influence of Preoperative Anticoagulant and Antiplatelet Therapy on Rebleeding Rates in Patients Suffering from Spinal Metastatic Cancer: A Retrospective Cohort Study"

_cancers, 2024, doi:10.3390/cancers16112052_

Round 1

Reviewer 1 Report

Comments and Suggestions for Authors

I am a spine surgeon. As a surgical provider, I am interested in the benefits and risks of spine surgery for metastatic malignancies presenting with paraplegia and pathological fractures.

I also sympathize with the theme of this issue, the impact of anticoagulants on outcomes and complications, which is also an important issue to be discussed clinically. I also agree with the results that the adverse effects of perioperative bleeding-related complications are not so high when anticoagulants are taken.

I would like to make some comments on this manuscript.

1. Regarding the type of surgery, is it standard practice to analyze a wide variety of surgeries together? I was uncomfortable with the idea of evaluating vertebroplasty or biopsy and discectomy together. (I feel that they are different procedures (both in terms of surgical invasiveness and bleeding risk). Also, regarding fusion surgery, the amount of bleeding should be different between open and percutaneous surgery. I was disappointed that this point was not taken into consideration in this study.

2. Regarding anticoagulants, a variety of drugs such as antiplatelet agents and DOACs were mentioned, and many cases of patients taking oral antiplatelet agents were included in the present study. Although there were not enough cases to compare the differences between drugs, I believe that some previous studies have examined the differences between drugs in spine surgery and fracture surgery. I thought it would be good to have a discussion on this point as well. (I am not sure if we can conclude that all anticoagulants are acceptable in general.)

3. the shorter the surgical site, the more likely it is that posterior hemorrhage (which by definition is an increase in paralysis and pain due to the epidural hematoma, right?) I was as surprised as the author about the result of this study, which showed that posterior hemorrhage (by definition, paralysis and increased pain due to epidural hematoma, right? Is there any difference in the method of hemostasis, whether or not a drain is placed, etc.? The more free space between bone and soft tissue, the less the epidural hematoma may compress the nerve tissue, but are there any previous studies similar to these results?

In addition, the status of approval by the Ethics Committee should also be included.

Author Response

Response to reviewer 1 comments

Thank you very much for your efforts and for the time to review this manuscript. We will try to respond to your comments in detail in the following point-by-point list:

Comments 1:

Regarding the type of surgery, is it standard practice to analyze a wide variety of surgeries together? I was uncomfortable with the idea of evaluating vertebroplasty or biopsy and discectomy together. I feel that they are different procedures (both in terms of surgical invasiveness and bleeding risk). Also, regarding fusion surgery, the amount of bleeding should be different between open and percutaneous surgery. I was disappointed that this point was not taken into consideration in this study.

Response 1:

In our work, we primarily focus on the underlying disease of spinal metastasis and the surgical interventions required for its treatment. As you are no doubt aware, the indication for surgical treatment and the type of operation vary depending on clinical presentation and imaging findings. Specifically, compression of neuronal structures or instability caused by spinal tumors necessitating surgical intervention can be addressed through various techniques. In our study, we aimed to evaluate all surgical methods employed for this indication and assess their associated risks of bleeding or rebleeding. Therefore, we did not restrict our analysis to any specific surgical technique. Our objective was to identify statistically significant differences in postoperative bleeding risks among the different surgical methods. The described decompression procedures were only dorsal decompressions of the neural structures, which were compressed by tumor mass. Interestingly, in our cohort, there was no significant difference in bleeding risks between the various surgical modes. This finding contrasts with existing literature, which typically reports higher bleeding rates for open surgeries compared to minimally invasive procedures. We have also to mention, that in our cohort there was no solely performed discectomy. Despite this, our data did not corroborate these differences, underscoring the importance of individualized surgical planning and comprehensive risk assessment to optimize patient outcomes.

However, we would like to thank you for your valuable contribution and have added the following passage to our manuscript for better clarification:

We aimed to evaluate all surgical methods employed for the treatment of spinal metastases and assess their associated risks of bleeding or rebleeding depending on the intake of anticoagulant medication. Consequently, our analysis did not restrict itself to any specific surgical technique. Our objective was to identify statistically significant differences in postoperative bleeding risks among the different surgical methods” (Row 235-239, marked green)

Comments 2:

Regarding anticoagulants, a variety of drugs such as antiplatelet agents and DOACs were mentioned, and many cases of patients taking oral antiplatelet agents were included in the present study. Although there were not enough cases to compare the differences between drugs, I believe that some previous studies have examined the differences between drugs in spine surgery and fracture surgery. I thought it would be good to have a discussion on this point as well. (I am not sure if we can conclude that all anticoagulants are acceptable in general.)

Response 2:

Thank you for this valuable comment. In response to your suggestion, we have broadened our literature search to include the additional references you recommended. The updated manuscript, which has been uploaded for your review, now includes an additional paragraph in the discussion:

“While standardized guidelines exist for discontinuing these drugs, data on urgent cases is sparse. For instance, patients with coronary heart disease on aspirin or dual therapy with clopidogrel face higher intraoperative bleeding risks. However, a cohort study by Cuellar et al. found no increased bleeding complications in surgeries with aspirin alone. (21) Despite the rise in direct oral anticoagulant (DOAC) use, specific perioperative guidelines for spine surgery are lacking. Dabigatran shows similar perioperative bleeding and thrombotic risks compared to warfarin, with procedures often feasible within 48 hours of discontinuation. (22) Short preoperative discontinuation (<24 hours) of DOACs may be acceptable, albeit with a slightly increased need for perioperative PRBC transfusion (23). a careful risk assessment is essential before surgery, especially for those with a more complex history of antiplatelet/anticoagulation therapy.” (Row 200-210, marked yellow)

Comments 3:

The shorter the surgical site, the more likely it is that posterior hemorrhage (which by definition is an increase in paralysis and pain due to the epidural hematoma, right?) I was as surprised as the author about the result of this study, which showed that posterior hemorrhage (by definition, paralysis and increased pain due to epidural hematoma, right? Is there any difference in the method of hemostasis, whether or not a drain is placed, etc.? The more free space between bone and soft tissue, the less the epidural hematoma may compress the nerve tissue, but are there any previous studies similar to these results?

Response 3:

Thank you for your comment. The principles of hemostasis are generally consistent across different types of surgeries. We were intrigued by the unexpected result from our statistical analysis regarding the increased risk on postoperative hematomas in operations with fewer levels. In response, we conducted a thorough literature review focused on this specific topic. Surprisingly, we found that existing literature predominantly addresses the occurrence of symptomatic epidural hematomas in less invasive operations following endoscopic surgery. However, the presence of residual tumor masses or suboptimal postoperative blood pressure management could potentially increase the risk of secondary bleeding in patients undergoing surgery for spinal metastases. For example, as we have already mentioned in our discussion, patients undergoing major spinal surgery who are monitored postoperatively in an intensive care unit may experience better blood pressure management. However, the use of drainage showed no significant influence on the reduced incidence of symptomatic secondary bleeding.

Additionally the ethics committee approval was added to the form.

Reviewer 2 Report

Comments and Suggestions for Authors

I had the privilege to review the manuscript entitled "The influence of preoperative anticoagulant and antiplatelet therapy on rebleeding rates in patients suffering from spinal metastatic cancer: a retrospective cohort study" submitted by Urban et al.

The authors reviewed their personal experience with anticoagulant and anti platelets medications in patient treated with surgery for spinal tumours. 

They conclude that surgical intervention for spinal metastatic cancer appears to be safe regardless of the patient's preoperative anticoagulation status. However, it is important to customize preoperative planning an preparation for each patient.

In my opinion this manuscript is well structured and designed, although this topic has been discussed in the past. Once again, no definitive guide lines can be obtained but we do get more supportive data on the perioperative risks for these patients.

I found the material and methods legit and the results appropriate.

I do find the tables clear and helpful.

In my opinion, this manuscript can be published in the present form.

Author Response

Thank you very much for your efforts and for the time to review this manuscript. We were very pleased that they were so appreciative of our work.